# Comparative Analysis of Molecular Functions and Biological Role of Proteins from Cell-Free DNA-Protein Complexes Circulating in Plasma of Healthy Females and Breast Cancer Patients

**DOI:** 10.3390/ijms24087279

**Published:** 2023-04-14

**Authors:** Oleg Tutanov, Aleksei Shefer, Yuri Tsentalovich, Svetlana Tamkovich

**Affiliations:** 1V. Zelman Institute for Medicine and Psychology, Novosibirsk State University, 630090 Novosibirsk, Russia; ostutanov@gmail.com (O.T.);; 2International Tomography Center, Siberian Branch of Russian Academy of Sciences, 630090 Novosibirsk, Russia; yura@tomo.nsc.ru

**Keywords:** histones, circulating DNA, nucleoprotein complexes, plasma, MALDI-TOF mass spectrometry, breast cancer

## Abstract

Cell-free DNA (cfDNA) circulates in the bloodstream packed in membrane-coated structures (such as apoptotic bodies) or bound to proteins. To identify proteins involved in the formation of deoxyribonucleoprotein complexes circulating in the blood, native complexes were isolated using affinity chromatography with immobilized polyclonal anti-histone antibodies from plasma of healthy females (HFs) and breast cancer patients (BCPs). It was found that the nucleoprotein complexes (NPCs) from HF plasma samples contained shorter DNA fragments (~180 bp) than BCP NPCs. However, the share of DNA in the NPCs from cfDNA in blood plasma in HFs and BCPs did not differ significantly, as well as the share of NPC protein from blood plasma total protein. Proteins were separated by SDS-PAGE and identified by MALDI-TOF mass spectrometry. Bioinformatic analysis showed that in the presence of a malignant tumor, the proportion of proteins involved in ion channels, protein binding, transport, and signal transduction increased in the composition of blood-circulating NPCs. Moreover, 58 (35%) proteins are differentially expressed in a number of malignant neoplasms in the NPCs of BCPs. Identified NPC proteins from BCP blood can be recommended for further testing as breast cancer diagnostic/prognostic biomarkers or as being useful in developing gene-targeted therapy approaches.

## 1. Introduction

The presence of free circulating nucleic acids in the blood is currently beyond doubt [1]. Current research on cell-free DNA (cfDNA) focused on identifying combinations of markers (point mutations, microsatellite instability, amplification/deletions, epigenetic changes) to develop sensitive and accurate diagnostic tools, and to monitor the effectiveness of anticancer therapy [2,3]. However, despite numerous attempts to develop diagnostic systems based on “liquid biopsy”, so far, only one test based on the analysis of cfDNA methylation is currently approved by the FDA and used for colon cancer diagnosis (www.epiprocolon.com, www.epigenomics.com accessed on 10 November 2022). One of the most serious obstacles to the widespread implementation of such tests in applied oncology is the unsatisfactory signal-to-noise ratio [4]. To improve the sensitivity and specificity of “liquid biopsy” approaches, knowledge of the mechanisms of blood-circulating cfDNA origins and its further clearance, forms of cfDNA circulation, its primary structure, and the contribution of different forms of circulation to the total pool is required.

To date, it is known that blood cfDNA is circulating predominantly in the form of nucleoprotein complexes (NPCs), including complexes with histones [5,6], amyloid P [7], and blood and cell proteins [8,9], as well as in apoptotic particles [10,11,12]. The structure, properties, and biological activity of NPCs, the features of their circulation and protein composition are currently not well understood. However, a number of studies indicate that functionally active DNA can be transported to other cells via cNPCs, that cfDNA is necessary for the normal functioning of multicellular organisms, and that it can be associated with the development of a number of pathological processes [13,14].

Indeed, the development of oncological diseases was shown to be accompanied by an increase in the concentration of cfDNA in the blood [15], but the causes, mechanisms, and consequences of this phenomenon are still unknown. In addition, the alterations found in the cfDNA present in the bloodstream of cancer patients are consistent with those observed in the DNA of their corresponding tumor cells [2,3,16], and proteins with DNA-binding motifs were shown to protect this DNA from hydrolysis by endogenous nucleases [17], forming NPCs [5,6,7,8,9]. Thus, a comparative analysis of proteins in the NPCs circulating in the blood of healthy donors and cancer patients might allow for the expansion of the fundamental knowledge of the mechanisms that lead to the appearance of extracellular nucleic acids in the external environment, ensure their circulation in the blood and, as a consequence, reveal the mechanisms of tumor dissemination at the molecular level.

In the current study, we isolated native deoxyribonucleoprotein complexes circulating in the blood plasma of healthy females (HFs) and primary breast cancer patients (BCPs), identified NPC proteins, and performed bioinformatics analysis.

## 2. Results

### 2.1. Concentration of Histones and cfDNA in Plasma of HFs and Untreated BCPs

The concentration of histones H2a and H4 in the plasma of HFs (n = 30) and BCPs (n = 30) was estimated by ELISA, with a sensitivity of 124 pg/mL and 56 pg/mL, respectively. Both sets were characterized with coefficients of variation less than 10%. Histone H2a was detected in 80% of BCP plasma samples with a median concentration of 498 pg/mL of blood. In the blood plasma of HFs, this protein was found in 60% of samples with a median concentration of 149.2 pg/mL (Figure 1a). The revealed difference in the concentration of histone H2a in the blood of the control group and cancer patients was significant (*p* = 0.039, Mann–Whitney U test) and consistent with the literature data [18,19,20]. Since histone H2a is a part of circulating NPCs, and it is known that the cfDNA concentration in the cancer patients blood is significantly increased [15,16], the theoretically predicted increase in the concentration of histone H2a as a component of NPCs was confirmed using sandwich ELISA. At the same time, the concentration of H4 histone in both HF and BCP plasma samples was lower than the detection level of the method. The obtained results indirectly indicate that histone H4 is not available for binding by antibodies in the NPC structure.

The cfDNA concentration in the plasma of HFs (n = 30) and BCPs (n = 30) was estimated by qPCR specific for LINE-1. A significant increase in the cfDNA concentration was found for BCPs compared to HFs (median 29 versus 8 ng/mL of total blood, *p* = 0.00008, Mann–Whitney U test) (Figure 1b). The revealed difference between the tumor patients and the controls coincides with previous studies [16,21,22,23].

### 2.2. Characterization of NPCs Circulating in the Blood Plasma of HFs and BCPs

We used affinity chromatography with immobilized anti-histone antibodies to identify proteins in histone-containing NPCs involved in cfDNA homeostasis.

The size of DNA in isolated NPC samples was assessed using capillary electrophoresis after DNA isolation with a commercial “BPD100” kit (BioSilica Ltd., Novosibirsk, Russia). Since the cfDNA concentration in HF blood plasma was quite low, individual samples of DNA from NPCs were pooled, whereas the samples of DNA obtained from BCP plasma NPCs were analyzed individually (only for cases with high plasma cfDNA concentration).

We also found that NPCs from HF plasma samples contained more short DNA fragments (~180 bp) (Figure 2a) than NPCs from BCP plasma (Figure 2b,c).

It was shown that the share of DNA in the NPCs from cfDNA in blood plasma in HFs and BCPs did not differ significantly (the median was 26% and 30%, respectively, *p* = 0.655, Mann–Whitney U-test) (Figure 3a), as well as the share of protein in histone-containing NPCs from blood plasma total protein (the median was 3.00% and 3.05%, respectively, *p* = 0.521, Mann–Whitney U-test) (Figure 3b).

### 2.3. Annotation of Identified NPC Proteins

Using MALDI-TOF mass spectrometry, 177 and 169 proteins were identified with high reliability (*p* < 0.05) in the NPCs of HFs and BCPs, respectively (Appendix A). Of these, only 12% (38 proteins) were common to both groups (Appendix A).

A comparison of the 308 identified NPCs proteins with extracellular vesicle proteins annotated in the ExoCarta database (http://exocarta.org accessed on 1 March 2023) revealed a match with four proteins: MIL1 (UniProt ID Q9BXK5), COPS9 (UniProt ID Q9BX79), HAND1 (UniProt ID O96004), and TCP1 (UniProt ID Q99832), indirectly indicating the absence of any extraneous proteins that are not components of the NPCs.

It should be noted that the most commonly represented 12 universal proteins (Homeobox protein Hox-C5, Probable G-protein-coupled receptor 22, Putative G antigen family D member 1, Transcription factor jun-D, Translocating chain-associated membrane protein 1-like 1, Myosin regulatory light chain 12A, EF-hand calcium-binding domain-containing protein 9, Splicing factor U2AF 35 kDa subunit, Insulin-degrading enzyme, Spermatogenesis-associated protein 3, cAMP-responsive element modulator, Beta-1,3-glucosyltransferase) were found in 76% of the samples.

Then, using STRING web tool PPI network for 38 NCP universal proteins was constructed (Figure 4). It was found that GUCY1A3, MYL12A, MYL9, KRIT1, APBB1P, RALA, MAP3K14, and SPG7 displayed the majority of protein–protein interactions (PPI).

Histones H2a, H2b, and H3 were included in the analysis but were not confirmed by MALDI-TOF due to their high Lys contents and consequent excessive trypsinolysis.

Of note, CFHR2 (Complement factor H-related protein 1)—one of the proteins of HF NPCs was shown in previous studies to play a major role in the degradation of blood-circulating cfDNA along with FSAP and DNase I [11,24,25]. The CFHR2 protein identified in the current study is able to bind to the substrate, blocking factor H binding sites [26], thus increasing the duration of cfDNA circulation in the NPC. Moreover, H2AJ and HAT1 proteins were identified in HF blood NPCs. These proteins are known to facilitate disopsonin functions, thus increasing the circulation time of cfDNA and reducing the rate of its uptake by liver cells [27]. In addition, HMGB4 protein was identified in HF plasma NPCs. HMGB4 contains the HMGB domain (high mobility group box domain) allowing for the binding of DNA and other proteins. In HMGB1, this domain was shown to be involved in the binding of cfDNA, TLR2, and RAGE proteins mediating cfDNA internalization [28,29,30,31]. Currently, the HMGB4 protein is relatively poorly described and studied; however, its structural similarities might indicate that, like HMGB1, it can not only bind cfDNA [32], but also regulate its internalization, intercellular signaling, and immunostimulatory properties.

BCP plasma NPCs were significantly enriched with unique proteins containing an ion transport domain (IPR005821) and categories GO “ion transport” (GO:0006811) and “transmembrane transport” (GO:0055085) (P48995.1, Q6PIU1.2, Q8NEC5.3, Q96KK3.2), as well as unique proteins that are voltage-gated ion channels (IPR027359) (Q6PIU1.2, Q8NEC5.3, Q96KK3.2). These findings suggest the significance of these four proteins in the secretion of NPCs by donor cells and the uptake of circulating NPCs by recipient cells.

Functional enrichment analysis was performed for all NPC proteins from blood plasma of HFs and BCPs using STRING software (www.string-db.org, accessed on 1 March 2023). The obtained lists of enriched 46 cellular components, 57 molecular functions, and 144 biological processes GO terms with corresponding *p*-values were clustered and visualized with heat maps using R.

The cellular component GO terms universally enriched for NPCs included “intracellular organelle”, “macromolecular complex”, and “nucleus”. Notably, NPC proteins from HF blood displayed enriched GO terms with lg *p* > 5, such as “protein-DNA complex”, terms associated with the replication fork (“replisome”, “nuclear replisome”, “nuclear replication fork”, and “replication fork”), and terms associated with chromosomes (“chromosomal part”, “nuclear chromosomal part”, “chromosome”, “nuclear chromosome”). NPC proteins from the blood plasma of BCPs were uniquely enriched for “protein complex”, terms associated with neurons (“neuron spine”, “dendritic spine”, “dendrite”, and “neuron projection”), and terms associated with the cytoskeleton (“myosin complex”, “interphase microtubule organizing center”, “myosin I complex”, “myosin II complex”,, etc.) (Figure 5). Thus, nuclear and organelle proteins, as well as proteins participating in macromolecular complexes, are widely represented among blood NPC proteins, both in normal and cancer donors’ blood. Blood NPCs of HFs were uniquely enriched in proteins that directly interact with DNA (including those in protein–DNA complexes), while blood NPCs of BCPs were uniquely enriched in cytoskeleton and protein complex proteins.

Moreover, the molecular function GO analysis revealed that NPC proteins were commonly enriched for functions such as “DNA binding”, “rDNA binding”, “nucleic acid binding”, “sequence-specific DNA binding”, and “protein binding”, as well as enzymatic activities and receptor binding proteins (“pyrophosphatase activity”, “hydrolase activity, acting on acid anhydrides”, “G-protein coupled amine receptor activity”, “adrenergic receptor activity”) (Figure 6).

NPC proteins unique to HFs blood were enriched for terms associated with receptor binding (“alpha-2A adrenergic receptor binding”, “adrenergic receptor binding”, “hepatocyte growth factor receptor binding”, “oxytocin receptor activity”, “G-protein coupled receptor binding”), as well as enzymatic activities (“intramolecular oxidoreductase activity”, “malate dehydrogenase activity”, “transferase activity”, “sulfotransferase activity”, “aldehyde-lyase activity”, “lyase activity”, “threonine aldolase activity”, “isomerase activity”).

Uniquely enriched GO molecular function terms of NPC proteins from BCP blood were “Wnt-activated receptor activity”, “dopamine binding”, “transmembrane signaling receptor activity”, “transcription factor activity, sequence-specific DNA binding”, “rRNA binding”, etc.

The Wnt signaling pathway includes a complex network of intracellular interactions. Its ligands are able to trigger at least three different signal transduction pathways: a canonical one and two non-canonical ones. The canonical Wnt signaling pathway is involved in the control of cell proliferation and differentiation. In contrast, non-canonical Wnt signaling pathways tend to affect cytoskeletal organization and cell motility. Currently, it is assumed that canonical and non-canonical Wnt signaling cascades affect different stages of tumor development (Figure 6).

Thus, the proteomic profiles of blood circulating NPCs in normal and breast cancer showed enrichment of DNA- and protein-binding proteins. In HF blood NPCs, there is a unique enrichment of receptor-binding proteins and proteins with enzymatic activities, while in BCP blood, there is enrichment of proteins with Wnt-activated receptor activity. The common proteomic profiles of NPCs from the blood of HFs and BCPs are enriched with terms related to biological processes such as “nitric oxide mediated signal transduction”, “NADP biosynthetic process”, “regulation of system process”, “vitamin biosynthetic process”, “coenzyme metabolic process”, “cellular process”, “positive regulation of metabolic process”, and “regulation of metabolic process”. Proteins from BCP NPCs are involved in biological processes associated with regulation, including matrix biosynthesis. Proteins unique to HF NPCs are enriched in “DNA replication initiation”, “malate metabolic process”, “positive regulation of multicellular organismal metabolic process”, “RNA metabolic process”, “regulation of exocyst assembly”, and “regulation of exocyst localization”, among others (Figure 7).

Therefore, the NPC proteins in the blood are universally enriched for proteins that regulate signal transduction and cellular metabolism (including interactions with nucleotides). Additionally, for HF blood NPCs, proteins involved in the regulation of exocyst assembly and localization, DNA replication, and RNA metabolism were uniquely enriched, while for BCP blood NPCs, proteins involved in the regulation of transcription and fusion with the plasma membrane were uniquely enriched.

### 2.4. Characterization of NPC Proteins Involved in Malignant Neoplasms Development

Using the dbDEPC 3.0 database (database of Differently Expressed Proteins in Human Cancer) [33], we found that 58 (35%) of the proteins in NPCs from BCPs were associated with malignant neoplasms, of which 22 (13%) were associated with the development of breast cancer (Figure 8). It is noteworthy that hyper-expressed proteins associated with breast cancer development predominated in NPCs (59% hyper-expressed vs. 36% hypo-expressed proteins) (Table 1).

It is currently unknown whether NPCs are specifically assembled when cfDNAs are released as complexes with nucleosomes and whether, like exosomes, they are the molecular imprint of the secreting cell, including overexpressed proteins as “passenger” proteins. However, regardless of the nature of the inclusion of these tumor-associated proteins in circulating NPCs, the data obtained may serve as a basis for future promising methods for their detection.

## 3. Discussion

Incubation of purified DNA samples with blood [34] and other types of biological fluids leads to rapid hydrolysis of the nucleic acids. Nevertheless, a certain concentration of cfDNA in blood is normally maintained [15]. One of the mechanisms ensuring the prolonged circulation of DNA in blood, apparently, may be the packing of nucleic material fragmented by apoptosis into apoptotic bodies surrounded by the cell membrane [35].

The Increased content of cancer-specific DNA sequences in the pool of cfDNA in blood [36], as well as increased cfDNA concentration in cell culture containing negligible amounts of apoptotic and/or necrotic cells [37], suggest that cfDNA can be actively secreted by cells as a part of NPCs. Indeed, in cervical adenocarcinoma cells (HeLa culture) and human umbilical vein endothelial cells (HUVEC culture), DNA accumulation in the culture medium was shown to persist during the lag phase and at the beginning of the exponential growth phase of cell cultures. The authors believe that this cannot indicate the apoptotic origin of DNA but can be explained by the presence of active DNA secretion mechanisms by cells [38]. Another group of authors [39] demonstrated a capability to actively secret DNA-containing complexes in response to bacterial infection (Streptococcus pyogenes) on HMC-1 mastocyte culture.

Another example is the active secretion of nucleosomes by neutrophils. After 30 min of co-incubation of human neutrophils with platelet-rich blood plasma and type I collagen preactivated by phorbol myristate acetate, an 11-fold increase in nucleosome concentration was observed in the culture medium [40]. Additionally, in a seven-cell line in vitro system, a statistically significant correlation was found between the level of glycolysis and the concentration of cfDNA. The cfDNA fragments in the culture media were represented by both apoptotic ladder and high molecular weight fragments with the most common length of ~2000 bp. During cell cultivation, the proportion of fragments with a length multiple to that of the nucleosome ladder decreased while the proportion of long DNA fragments increased, suggesting that cfDNA in the medium may not have originated from apoptotic or necrotic cells [41].

The protection of DNA from hydrolysis by blood nucleases can also be achieved by packaging nucleic acids into oligonucleosomal particles. The presence of nucleosomes in blood plasma was confirmed through plasma DNA electrophoresis in agarose/PAA gel [16,42,43] and histone-specific ELISA [6]. The ability of linker and core histones to directly pass through the plasma membrane [10] suggests that DNA-histone complexes are probably capable of carrying DNA across membranes inside the cell. It should be noted that nucleosomes containing histone H1 are insoluble in the physiological ionic medium (at a concentration of divalent cations of 3 mM) characteristic of blood plasma, but serum amyloid P is capable of replacing histone H1, thus increasing nucleosome solubility in blood [7]. In addition, the presence of the locus responsible for DNA binding to histone H3K27me2 within cfDNA seems to be necessary for export and stabilization of cfDNA [10].

In the current study, the elevated concentrations of histone H2a and cfDNA in BCP plasma compared with HFs are consistent with the literature data [7,15,16,22]. However, despite numerous scattered data on proteins capable of affinity binding nucleic acids [44], it is still unknown which proteins, except histones and amyloid P, are involved in the formation of circulating NPCs in the bloodstream. For this reason, affinity chromatography was used to identify the NPC proteins, allowing the isolation of native histone-containing deoxyribonuclein complexes. Bioinformatic analysis showed that HF blood NPCs are uniquely enriched for proteins that bind receptors and carry out enzymatic activity, and proteins involved in the regulation of exocyst assembly and localization, DNA replication, and RNA metabolism. In contrast, BCP blood NPCs are uniquely enriched for proteins with Wnt-activated receptor activity and proteins involved in the regulation of transcription and fusion with the plasma membrane. In addition, proteins associated with malignant cell rearrangement were detected in such complexes circulating in the blood of cancer patients, which indicates that part of the cfDNA is of tumorigenic origin. The presence of tumor-related proteins only in NPCs from BCPs blood demonstrates the potential of NPCs as a source of material for protein-based cancer diagnostics and potential enrichment of cancer-related cfDNA using specific antibodies.

## 4. Materials and Methods

### 4.1. Ethics Statement

Blood samples from HFs (n = 30, median age 50) were obtained from Novosibirsk Central Clinical Hospital. HFs did not have any female disorders (dysplasia, endometriosis, etc.) or any malignant diseases.

Blood samples from untreated BCPs (n = 30, median age 56) were obtained from the Novosibirsk Regional Oncology Dispensary. The clinicopathological parameters of the patients with breast cancer are presented in Table 2.

The subtype of breast cancer was established by the immunohistochemical study of tissue samples after surgery (expression of receptors for estrogen (ER) and progesterone (PR)), HER-2 status and the level of proliferative activity (expression of Ki67) in accordance with the St. Gallen Consensus Recommendation [45]. IHC was prepared as described [46]. For ER and PR expressions, the cases were classified as positive when nuclear immunoreactivity was in ≥1% of tumor cells according to the American Society of Clinical Oncology/College of American Pathologists (ASCO/CAP) guidelines [47]. Sections stained with ER and PR were scored using the H-score method [48]. HER2 protein-positive status was defined as a score of 3+ by IHC or 2+ by IHC together with the confirmed *c-erbB2* gene amplification by fluorescence in situ hybridization (FISH).

### 4.2. Blood Treatment

Venous blood (9 mL) was collected in K_3_EDTA spray-coated vacutainers (Improvacuter, China, cat. No. 694091210), immediately mixed using a rotary mixer, placed at +4 °C, and fractionated into plasma and blood cells within an hour after blood sampling. Blood was centrifuged at 290× *g* for 20 min. Blood plasma was then transferred into a new tube and centrifuged a second time at 1200× *g* for 20 min. Plasma samples were stored at −80 °C in aliquots and defrosted before assaying.

Before subsequent manipulations, all samples were tested for the absence of hemolysis/lysis of blood cells by assessing the level of hemoglobin (the absorbance of <0.175 at 414 nm). Blood samples with signs of hemolysis that occurred in cancer patients due to the general disturbance of lipid metabolism were excluded from the study.

### 4.3. Plasma Histones Measurment

The concentration of Histone H2a and Histone H4 in plasma samples of HFs and BCPs was measured using ELISA kit developed by Cloud-Clone Corp. (Wuhan, Hubei, China) according to the manufacturer’s instructions. Before analysis, plasma samples were thawed at room temperature and centrifuged at 1500× *g* for 10 min, 10 μL of plasma was used in assay. A microplate reader (Thermo Scientific Multiskan FC, Shanghai, China) was used to determine sample absorbance at 450 nm. The H2a histone ELISA sensitivity was 0.156–10 ng/mL; the H4 histone ELISA sensitivity was 0.078–5 ng/mL. Both assays were characterized by coefficients of variation less than 10%.The H2a or H4 measurement was set to 0 ng/mL when the absorbance of a patient sample was lower than the buffer blank. All measurements were executed in duplicates and results are expressed as ng/mL of blood.

### 4.4. The Synthesis of Sorbent with Immobilized Anti-Histone Antibodies and NPC Isolation by Affine Chromatography

Affine sorbent with immobilized anti-histone antibodies was synthesized from bromocyan-activated Sepharose [49]. Before starting the synthesis, the rabbit polyclonal anti-H2A; (PAQ850Hu01), anti-H2B; (PAQ006Hu01), and anti-H3 (PAA285Mi01) antibodies were mixed together (1 mg each, Cloud-Clone Corp. (Wuhan, Hubei, China)). A total protein (3 mg) was added to activated CL-4B Sepharose (3 g), adjusted pH to 9.5 with 0.5 M carbonate buffer, and incubated for 1 h under slow stirring. The remaining reaction centers were blocked by the addition of 0.5 M glycine buffer (pH 9.6, 1/10 of the volume) followed by incubation for 30 min under slow stirring. The resulting affine sorbent (3 mL) was successively washed on a glass filter with PBS (100 mL), 0.1 M glycine buffer containing 0.5 M NaCl (pH 2.5, 60 mL), 0.5 M borate buffer (pH 8.6, 60 mL), and PBS (100 mL). The sorbent was transferred to a chromatographic column (1 × 3.8 cm) and washed successively with PBS, glycine buffer, borate buffer, and PBS.

For the isolation of histone-containing NPC by affine chromatography, plasma (0.8 mL) was loaded onto the sorbent and incubated for 1 h at 4 °C and the sample was loaded again. The column was washed with PBS containing 5 mM EDTA (20 mL); then, it was washed with PBS containing 5 mM EDTA and 0.05% Tween-20 (20 mL), and again PBS containing 5 mM EDTA (20 mL). NPCs were eluted from the column in the opposite direction with the glycine buffer at a rate of 1 mL/min. The eluate was neutralized with the borate buffer (0.2 mL per 1 mL of the eluate). After affine chromatography the sorbent was regenerated by successive elution with PBS/EDTA, PBS/EDTA with 0.05% Tween-20, and PBS/EDTA (20 mL of each). The NPC samples were concentrated on Centricon 3 kDa filters for 4 h at 4000× *g*, +4 °C.

### 4.5. Plasma DNA and DNA from NPCs Isolation and Quantification

The DNA from plasma and from histone-containing NPC was isolated using the “DNA Isolation Kit” (BioSilica Ltd., Novosibirsk, Russia) according to the manufacturer’s protocols and concentrated by precipitation in acetone as triethylammonium salts: glycogen (5 μL) (Fermentas) and 50 mM trimethylamine (10 μL) were added to the sample (100 μL) and precipitated with acetone (500 μL). After incubation for 15 h at −20 °C, the DNA was precipitated by centrifugation at 17,000× *g* for 20 min at 4 °C and the precipitate was dissolved in H_2_O (12 μL). The concentration of isolated DNA was measured by quantitative polymerase chain reaction (Q-PCR) specific for long interspersed nuclear element 1 (LINE-1) repetitive elements [14]. The Q-PCR was performed with an ICycler iQ5 (Bio-Rad, Hercules, CA, USA) as described earlier [14]. Genomic DNA from human leukocytes served as a standard for obtaining the calibration curves. The DNA concentration was estimated according to the initial volume of each blood sample.

The size of NPC DNA was evaluated using a “High Sensitivity DNA Kit” and Agilent 2100 Bioanalyzer^TM^ (Agilent Technologies, Waldbronn, Germany) in SB RAS Genomics Core Facility (ICBFM SB RAS, Novosibirsk, Russia).

### 4.6. MALDI–TOF Mass Spectrometry Analysis

To evaluate the NPC protein concentration, a NanoOrange Protein Quantitation kit (NanoOrange^®^ Protein Quantitation Kit, Molecular Probes, Eugene, OR, USA) was used in accordance with the manufacturer’s recommendations.

Individual plasma NPC samples were separated according to their molecular weight using 10% SDS disc-electrophoresis. The gels were stained by Coomassie Brilliant Blue R250 (Sigma, St. Louis, MO, USA). The PAAG fragments containing proteins under study were treated using the modified Rosenfeld method. Briefly, PAAG fragments with proteins were washed from Coomassie R250 and SDS with an aqueous solution containing 50% acetonitrile (C_2_H_3_N) and 0.1% trifluoroacetic acid (C_2_HF_3_O_2_). Proteins absorbed in the gel were reduced with 45 mM DTT in 0.2 M ammonium bicarbonate (NH₄HCO₃) at 60 °C for 30 min, followed by protein alkylation with 100 mM iodoacetamide in 0.2 M NH₄HCO₃, at room temperature for 30 min. The gel fragments were dehydrated in 100% C_2_H_3_N. A 0.2 mM trypsin solution (modified by reductive methylation) (Sigma, T6567, USA), in a mixture with 0.1 M NH₄HCO₃ and 5 μM CaCl_2_, was added to each gel fragment and incubated for 30 min at room temperature. Then, peptide extraction buffer (60 μL) was added to the gel fragments, and samples were incubated for 16–18 h at 37 °C. The peptide fragments of proteins extracted from the gel were concentrated and desalted using C18 ZipTips micro-columns (Millipore, Darmstadt, Germany) according to the manufacturer’s instructions. The peptide mixture was eluted from the micro-column, on a target of the device plate with the saturated matrix solution.

Mass-spectra were registered at the Center of Collective Use “Mass spectrometric investigations” SB RAS on an Ultraflex III MALDI-TOF/TOF mass spectrometer (BrukerDaltonics, Bremen, Germany) in positive mode, with the range 700–3000 Da, and with 2,5-dihydroxybenzoic acid as a matrix. Proteins were identified by searching for appropriate candidates in annotated NCBI and SwissProt databases using Mascot software (Matrix Science Ltd., London, UK, www.matrixscience.com/search_form_select.html, accessed on 10 February 2023). The following parameters were used for searches: the acceptable mass deviation of the charged peptide (50 ppm)—0.05 Da; the acceptable number of missed cleavage sites—2; carbamidomethylation of cysteine residues was chosen as a fixed modification and the presence of oxidized methionine residues was chosen as a variable modification; identification reliability not lower than 95%.

### 4.7. Bioinformatics and Gene Ontology (GO) Analysis of NPC Proteins

GO functional enrichment analysis of all fractions was performed using STRING software (www.string-db.org, accessed on 1 March 2023) with the base settings for multiple proteins analysis. The redundancy reduction in the obtained data was conducted using REVIGO software (https://www.irb.hr/, accessed on 1 March 2023) with the allowed similarity of <0.5.

Heatmap plotting of the enrichment results obtained through STRING was conducted via R (https://www.R-project.org, accessed on 1 March 2023), using packages gplots, dendextend, colorspace.

Profiling of the proteins differently expressed during the development of various malignant diseases was performed using the dbDEPC 3.0 database [33].

### 4.8. Statistical Analysis

Statistical calculations were performed using Statistica 6.0 software. All data were expressed either as the median with interquartile ranges or as means with standard errors. To evaluate the differences, the Mann–Whitney U-test was performed. *p* < 0.05 was considered statistically significant.

## 5. Conclusions

The unique data obtained in the current study on the NPC proteins circulating in the bloodstream expand fundamental knowledge of the molecular mechanisms that ensure the transfer of genetic information and signals between cells. The knowledge about the unique proteins in NPCs of BCPs can be useful for the development of non-invasive methods of tumor diagnostics at preclinical stages, disease prognosis, and the development of technology for obtaining vectors for gene therapy based on natural transport system elements.

## Figures and Tables

**Figure 1 ijms-24-07279-f001:**
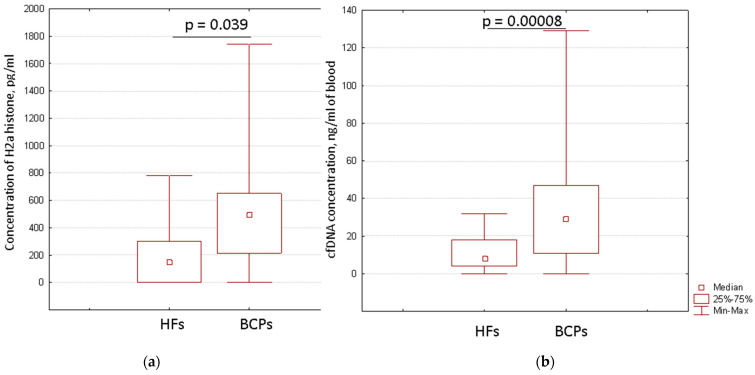
Concentration of histone H2a (**a**) and cfDNA (**b**) from the blood plasma of HFs and BCPs. Tukey box plots, median concentration with 25–75% and non-outlier range bars are indicated.

**Figure 2 ijms-24-07279-f002:**
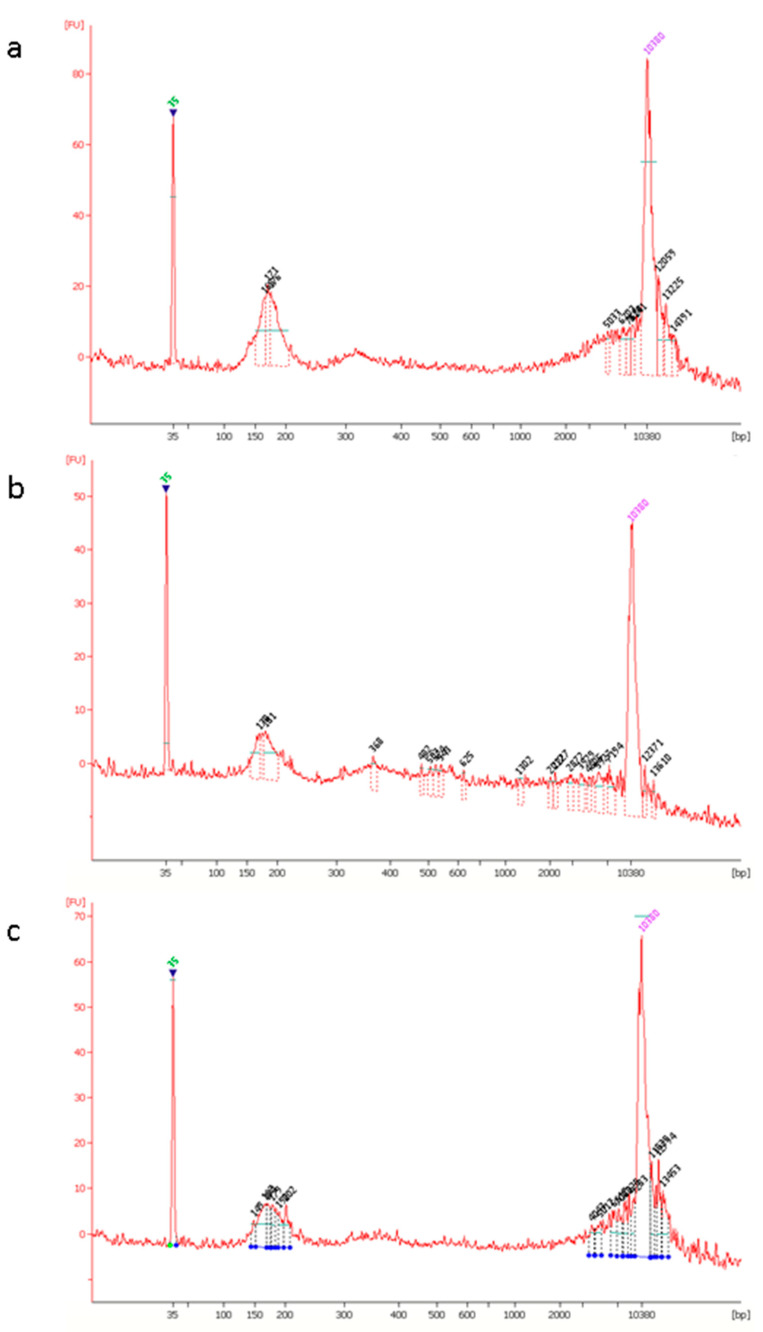
Electrophoregrams of pooled DNA sample from HF blood plasma NPCs (**a**) and individual DNA samples from BCP blood plasma NPCs (**b**,**c**). Data of Agilent 2100 Bioanalyser^TM^ assay. Positions of 35 and 10,380 bp markers are shown with arrows.

**Figure 3 ijms-24-07279-f003:**
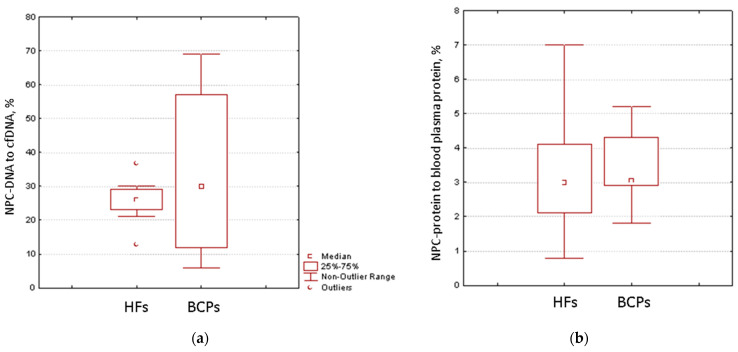
Characterization of NPCs circulating in the blood plasma of HFs and BCPs in terms of nucleic and protein components: (**a**) the share of DNA in NPCs versus concentration of cfDNA in blood plasma; (**b**) the share of protein in NPC versus total blood plasma protein. Tukey box plots, median concentration with 25–75%, and non-outlier range bars are indicated.

**Figure 4 ijms-24-07279-f004:**
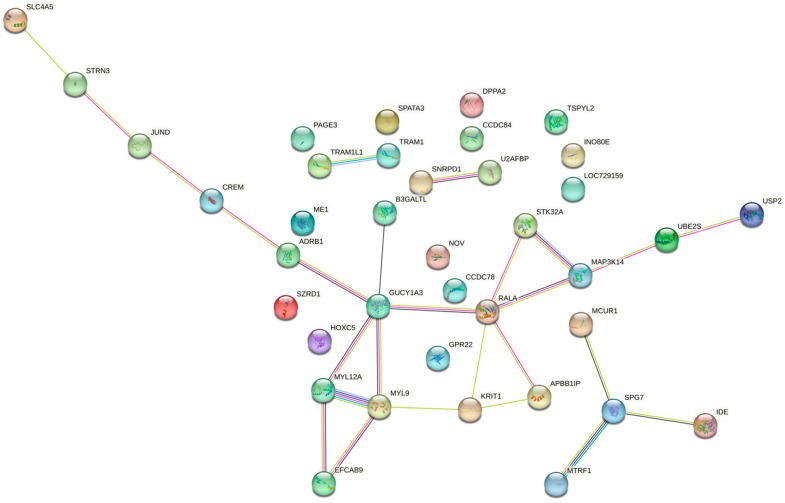
Protein–protein interaction network of 38 proteins that are universal to both BCPs and HFs blood-circulating NPCs. PPI networks plotted with STRING (http://string-db.org/, accessed on 1 March 2023) with following settings—minimum interaction score: high confidence (0.700); active interaction sources: textmining, experiments, databases, co-expression, neighborhood, gene fusion, co-occurrence.

**Figure 5 ijms-24-07279-f005:**
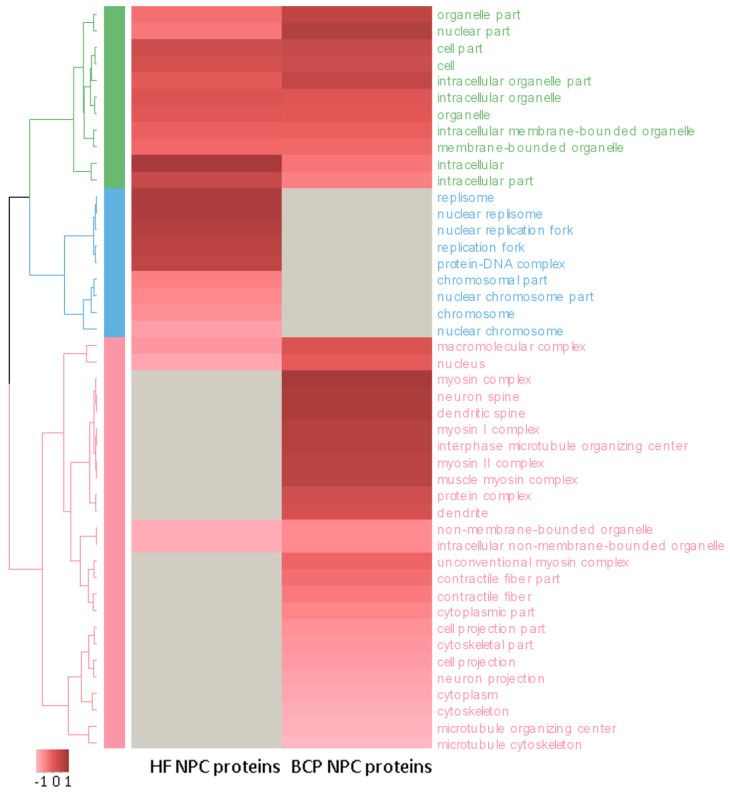
Heat map and dendrogram of the results of GO functional enrichment analysis of NPC proteins from BCPs and HFs by cellular components obtained via STRING via STRING (https://string-db.org/, accessed on 1 March 2023) and visualized using R (https://www.r-project.org/, accessed on 1 March 2023). GO results with *p* < 0.05 are shown as normalized −log (*p*) values.

**Figure 6 ijms-24-07279-f006:**
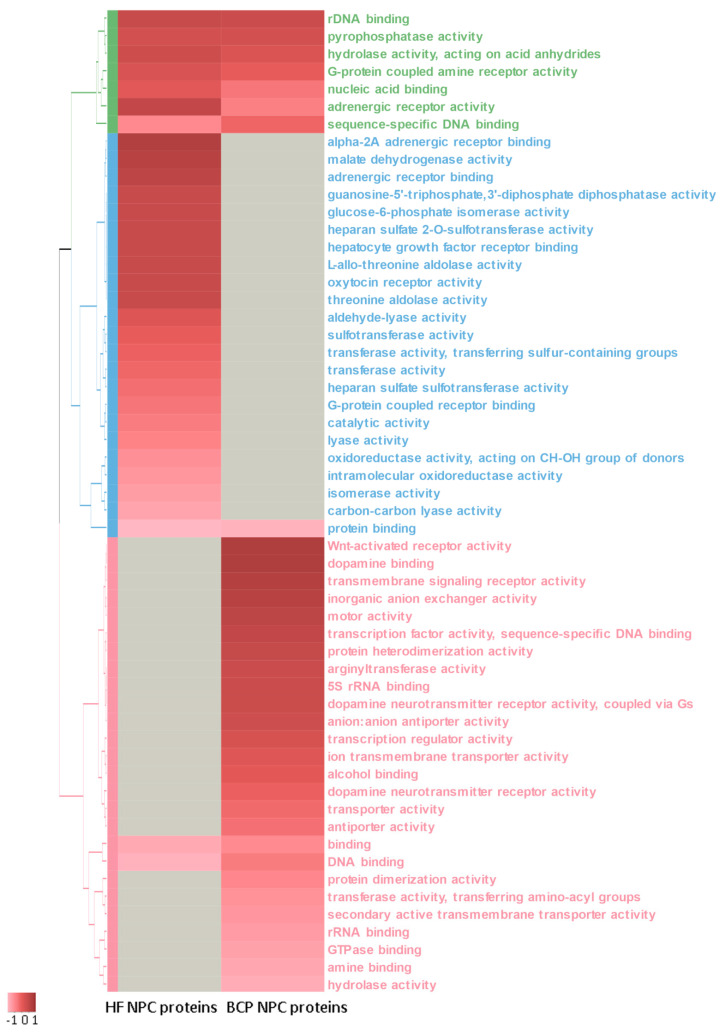
Heat map and dendrogram of the results of GO functional enrichment analysis of NPC proteins from BCPs and HFs by molecular functions obtained via STRING (https://string-db.org/, accessed on 1 March 2023) and visualized using R (https://www.r-project.org/, accessed on 1 March 2023). GO results with *p* < 0.05 are shown as normalized −log (*p*) values.

**Figure 7 ijms-24-07279-f007:**
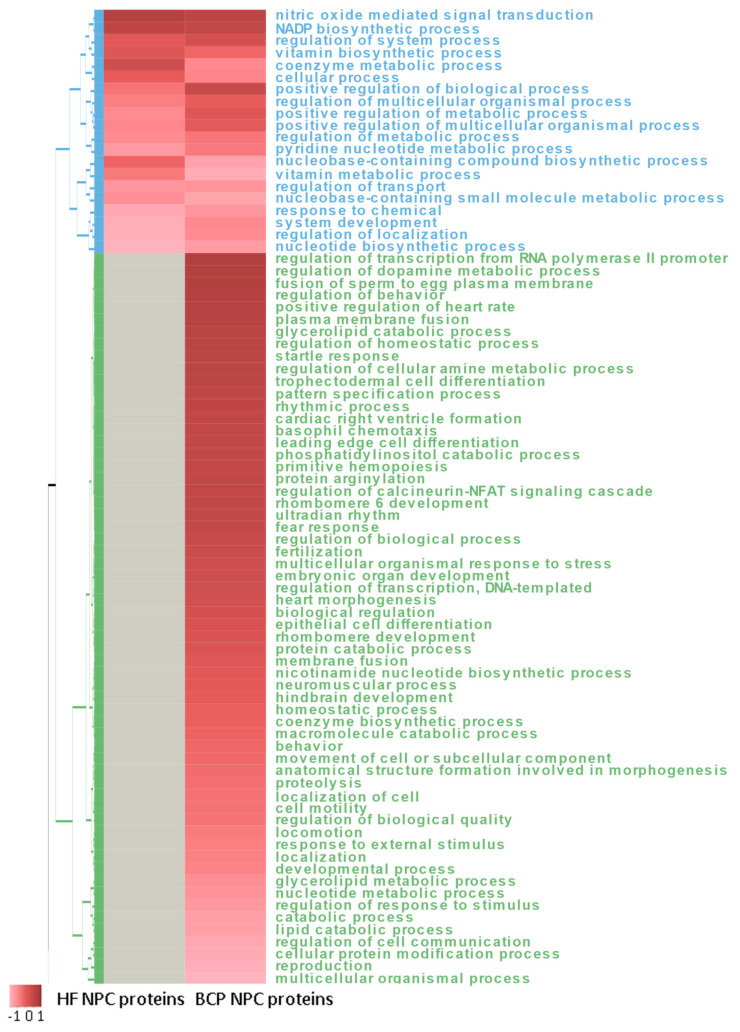
Heat map and dendrogram of the results of GO functional enrichment analysis of NPC proteins from BCPs and HFs by biological processes obtained via STRING (https://string-db.org/, accessed on 1 March 2023) and visualized using R (https://www.r-project.org/, accessed on 1 March 2023). GO results with *p* < 0.05 are shown as normalized −log (*p*) values.

**Figure 8 ijms-24-07279-f008:**
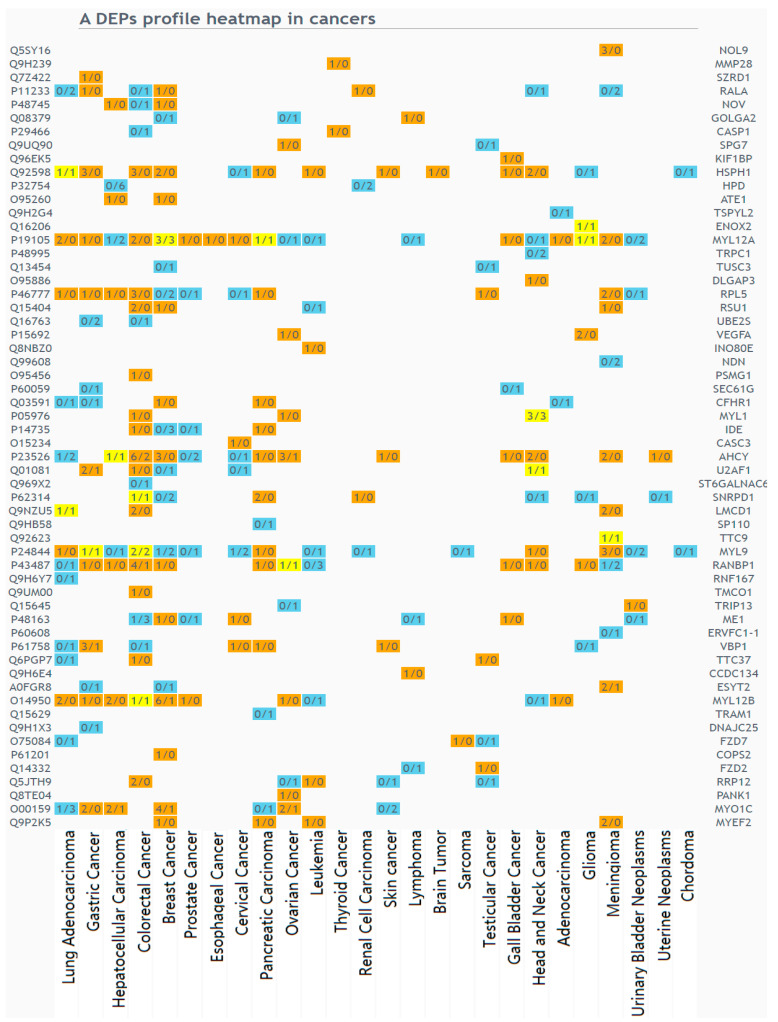
Heat map generated by dbDEPC 3.0 [33] from NPC proteins from BCP blood, associated with different cancer types. Orange means that the number of the studies identified this protein as up-regulated are more than the number of the studies identified the protein as down-regulated; blue means that the number of the studies identified this protein as up-regulated are less than the number of the studies identified protein as down-regulated; yellow means that the number of the studies identified the protein as up-regulated are equal to the number of the studies identified protein as down-regulated.

**Table 1 ijms-24-07279-t001:** Identified using dbDEPC 3.0, NPC proteins in the BCP blood, differentially expressed in breast cancer ^1^.

Protein Name	Gene Symbol	Protein Function	Expression
**Adenosylhomocysteinase**	**AHCY**	**Adenosylhomocysteinase. Has a key role in methylation control.**	Hyper-expressed ^2^
**Complement factor H-related protein 1**	**CFHR1**	**Complement factor H-related protein 1. Involved in complement regulation.**	
**COP9 signalosome complex subunit 2**	**COPS2**	**COP9 signalosome complex subunit 2. A key component of a complex involved in several intracellular processes, such as ubiquitination regulation, p53/TP53 phosphorylation, etc.**	
**Heat shock protein 105 kDa**	**HSPH1**	**Heat shock protein 105 kDa. Acts as a NEF factor for HSPA1A and HSPA1B. Prevents protein ugregation under stress.**	
**Myelin expression factor 2**	**MYEF2**	**Myelin expression factor 2. Myelin protein expression repressor.**	
**Myosin regulatory light chain 12B**	**MYL12B**	Myosin regulatory light chain 12B. Involved in cytokinesis, receptor capping and cell migration.	
**Unconventional myosin-Ic**	**MYO1C**	Myosin-Ic. Involved in intracellular transport.	
**Ran-specific GTPase-activating protein**	**RANBP1**	**Ran-specific GTPase-activating protein. May act in the intracellular signaling pathway, controlling the cell cycle, regulating, protein, and nucleic acid transport across the nuclear membrane**	
**60S ribosomal protein L5**	**RPL5**	**60S ribosomal protein L5. Ribosome component.**	
**Ras suppressor protein 1**	**RSU1**	**Ras suppressor protein 1. Possibly involved in Ras-pathway signal transduction.**	
Malic enzyme 1	ME1	NADP-dependent Malic enzyme 1. Involved in lipids methabolsm.	
Protein NOV homolog	NOV	Protein NOV homolog. Involved in a several cellular processes, including proliferation, adhesion, migration, differentiation, and surviving.	
Ras-related protein Ral-A	RALA	Ras-related protein Ral-A. Multifunctional GTPase, involved in a several cellular processes, such as gene expression, cell migration, cell proliferation, oncogenic transformation, and membrane transport.	
**Arginyl-tRNA-protein transferase 1**	**ATE1**	**Arginyl-tRNA-protein transferase 1. Involved in posttranslational conjugation of arginine with protein N-terminal aspartate or glutamate in ubiquitin pathway degradation.**	disputed status ^3^
**Golgin subfamily A member 2**	**GOLGA2**	**Golgin subfamily A member 2. Peripherical membrane of Golgi apparatus. Involved in vesicle binding and membrane fusion. Plays a key role in Golgi apparatus desorption in mitosis.**	
**Extended synaptotagmin-2**	**ESYT2**	**Extended synaptotagmin-2. Involved in EPR-cellular membrane contact generation. Plays key role in intracellular lipid transport and signal transduction.**	Hypo-expressed ^4^
**Myosin regulatory light chain 12A**	**MYL12A**	**Myosin regulatory light chain 12A. Myosin regulatory subunit, myosin subunit mediating contractile activity in both smooth muscle cells and nonmuscular cells. Participates in cytokinesis, receptor capping and cell migration**	
**Tumor suppressor candidate 3**	**TUSC3**	**Tumor suppressor candidate 3. Supporting N-oligosaccharil transferase.**	
Insulin-degrading enzyme	IDE	Insulin-degrading enzyme. Involved in insulin, glucagon, bradykinin, and other peptide cellular degradation. Acts in intercellular signal transduction.	
Myosin regulatory light polypeptide 9	MYL9	Myosin regulatory light polypeptide 9. Involved in cytokinesis, receptor capping, and cell migration.	
Small nuclear ribonucleoprotein Sm D1	SNRPD1	Small nuclear ribonucleoprotein Sm D1. Plays important role in splicing.	
U2 small nuclear RNA auxiliary factor 1	U2AF1	U2 small nuclear RNA auxiliary factor 1. Plays a key role in splicing. Mediates protein–protein and protein–RNA interactions.	

^1^ bold highlighted proteins unique to blood NPCs of BCPs; ^2^ number of studies that identified the protein as hyper-expressed > number of studies that identified the protein as hypo-expressed; ^3^ number of studies that identified the protein as hyper-expressed = number of studies that identified the protein as hypo-expressed; ^4^ number of studies that identified the protein as hyper-expressed < number of studies that identified the protein as hypo-expressed.

**Table 2 ijms-24-07279-t002:** Clinical characteristics of untreated BCPs.

		N (%)
Tumor stage	T1	21 (70%)
	T2	9 (30%)
Lymph node status	N0	24 (80%)
	N1	5 (17%)
	N2	1 (6%)
Distant metastasis	M0	30 (100%)
Receptor Status	ER-positive	30 (100%)
	PR-positive	30 (100%)
HER-2 Status	Negative	30 (100%)
Histologic Grade	II	26 (87)
	III	4 (13)
Histological type	Invasive ductal carcinoma	30 (100%)

## Data Availability

The data presented in this study are available on request from the corresponding author.

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
