# Peer review of "Comparative Analysis of Molecular Functions and Biological Role of Proteins from Cell-Free DNA-Protein Complexes Circulating in Plasma of Healthy Females and Breast Cancer Patients"

_ijms, 2023, doi:10.3390/ijms24087279_

Round 1
Reviewer 1 Report
Tutanov et al. presented very interesting results on blood circulating nucleoprotein complexes. NPCs from healthy females and breast cancer patients were compared and bioinformatics analysis reveals critical functions in these NPC proteins. A few comments are listed below.
The manuscript coupled high throughput analysis with bioinformatics. Have the authors performed validation of the molecules high lightened from the analysis?
Figure 6 and figure 7 have low resolution. The quality of the heatmap needs to be improved. Go terms font in figure 5 is different than that in figure 6 and 7.
Reviewer 2 Report
The topic of this study is interesting, and the study is well presented, well-articulated and thorough. The English is overall correct, and the structure of the text is logical and fluent. The conclusions, in my opinion, are reasonable and encourage more in-depth studies on the role of the cfDNA as a prognostic indicator in tumour settings. Identification of NPC's from breast cancer patient blood in tumor diagnostics at preclinical stages is a novel and up-and-coming strategy. My only concern for this study would be the sample size and its luminal A only representation. Increasing potentially the sample size and incorporating other subtypes would give more of diverse approach for the other breast cancer subtypes (albeit less frequent). Along these lines I suggest amending the article’s title. Also a stronger emphasis should be given on the comparative analysis in the discussion.
Reviewer 3 Report
The manuscript can be accepted after the authors correct the following comments:
1. The title unclearly and insufficiently reflects its content.
2. The abstract is inconsistent. In other words, the method should be inserted before the results and discussion. So, please rewrite the abstract section with this arranged: background, aim of study, methods, results and discussion, and finally the conclusion.
- Please note that the introduction part further paragraphs. Kindly, add new paragraphs at the introduction section with modern references.
4. Please make double check about the academic writing (needs native speakers in English).
5. The method section should be directly inserting after the introduction section.
6. The Result section should be directly inserting after the method section, then the discussion.
7. The resolution of manuscript figures is not clear. The figures need further clarification.
- The conclusion section should be reflecting the results in a good way.
- Make sure that all sentences are linked together.

Round 2
Reviewer 3 Report
Thanks for your corrections of the required comments. However, I am still wondering why the method section is after the results and discussion. I am pretty sure that the method section should be inserted before the results and discussion section, unless the journal style is required this arrangement.
Thank you